# High Parasitic Loads Quantified in Sylvatic *Triatoma melanica*, a Chagas Disease Vector

**DOI:** 10.3390/pathogens11121498

**Published:** 2022-12-08

**Authors:** Carolina Valença-Barbosa, Paula Finamore-Araujo, Otacílio Cruz Moreira, Marcus Vinicius Niz Alvarez, André Borges-Veloso, Silvia Ermelinda Barbosa, Liléia Diotaiuti, Rita de Cássia Moreira de Souza

**Affiliations:** 1Grupo Triatomíneos, Instituto René Rachou-Fiocruz Minas Gerais, Belo Horizonte, Minas Gerais 30190-002, Brazil; 2Plataforma de PCR em Tempo Real RPT09A, Laboratório de Virologia Molecular, Instituto Oswaldo Cruz/Fiocruz, Rio de Janeiro 21040-360, Brazil; 3Instituto de Biotecnologia de Botucatu, Universidade Estadual Paulista, São Paulo 18607-440, Brazil

**Keywords:** triatomine, *Trypanosoma cruzi*, *brasiliensis* complex, blood-feeding behavior

## Abstract

*Triatoma melanica* is a sylvatic vector species in Brazil. In We aimed to characterize the *Trypanosoma cruzi* discrete typing units (DTUs), the parasitic loads, and the blood meal sources of insects collected in rocky outcrops in rural areas in the state of Minas Gerais. An optical microscope (OM) and kDNA-PCR were used to examine natural infection by *T. cruzi*, and positive samples were genotyped by conventional multilocus PCR. Quantification of the *T. cruzi* load was performed using qPCR, and the blood meal sources were identified by Sanger sequencing the 12S rRNA gene. A total of 141 *T. melanica* were captured. Of these, ~55% (61/111) and ~91% (63/69) were positive by OM and KDNA-PCR, respectively. We genotyped ~89% (56/63) of the *T. cruzi*-positive triatomines, with TcI (~55%, 31/56) being the most prevalent DTU, followed by TcIII (~20%, 11/56) and TcII (~7%, 4/56). Only TcI+TcIII mixed infections were detected in 10 (~18%) specimens. A wide range of variation in the parasitic loads of *T. melanica* was observed, with an overall median value of 10^4^ parasites/intestine, with females having higher *T. cruzi* loads than N2, N4, and N5. TcII showed lower parasitic loads compared to TcI and TcIII. The OM positive diagnosis odds ratio between *T. cruzi* infection when the parasite load is 10^7^ compared to 10^3^ was approximately 29.1. The most frequent blood meal source was *Kerodon rupestris* (~58%), followed by *Thrichomys apereoides* (~18%), *Wiedomys cerradensis* (~8%), *Galactis cuja* (~8%) and *Gallus gallus* (~8%). Our findings characterize biological and epidemiological aspects of the sylvatic population of *T. melanica* in the study area, highlighting the need to extend surveillance and control to this vector.

## 1. Introduction

Chagas disease (ChD) is considered an enzootic disease transmitted by triatomine species and maintained in sylvatic mammals. It became an anthropozoonosis with the entry of humans into sylvatic environments, where *Trypanosoma cruzi,* the etiological agent of the disease, circulates [1]. Currently, ChD is a neglected disease and affects about 7 million people, with 25 million at risk in American countries [2]. The transmission of *T. cruzi* is highly complex, involving various species of both wild and domestic mammalian hosts and more than 150 triatomine species occupying different habitats [3,4,5,6]. The epidemiological importance of each triatomine species for human public health is related to their degree of involvement in domestic parasite transmission cycles.

In Brazil, advances have been made in the control of vector transmission, including the use of insecticide treatment for domiciles. Using such approaches, the elimination of the non-native species *Triatoma infestans* has been achieved throughout much of this national territory [7]. Although vector control measures have significantly reduced the risk of *T. cruzi* transmission, the existence of native triatomine species that occur in sylvatic environments represents a great challenge for public health since the elimination of these populations is extremely difficult, if not impossible [8].

To date, six different discreet typing units (DTUs) of *T. cruzi* have been described, the most recent being Tcbat [9,10]. Some DTUs have been correlated with different transmission cycles; TcI is predominant in sylvatic environments and preferentially associated with rodents, whereas TcII is frequently associated with domestic transmission cycles and linked with primates. The genotypes TcIII and TcIV have been mostly associated with sylvatic transmission cycles related to armadillos and raccoons, respectively, while TcV and TcVI are generally found in domestic cycles and are only rarely detected in wild reservoirs [1,11].

Despite each mammalian species playing a different role in the complex transmission network of *T. cruzi* in distinct habitats, particular DTUs are not exclusive to any species or even orders of mammals. Mammals that can act as a source for transmission of the parasite in an ecological system are referred to as “reservoir hosts.” Consequently, the preference of different triatomine species for blood-feeding on different mammalian species can influence the dynamics of ChD and their interaction (or not) with *T. cruzi* [12]. Thus, identifying the different *T. cruzi* lineages and their interaction with different mammalian hosts and triatomine vectors is potentially crucial for understanding their roles in different transmission cycles.

One important interaction between hosts and vectors is the development of *T. cruzi* in the triatomine gut content. According to Verly et al. [13], multiple variables influence the success of the host infection, including the parasitic load of the triatomine vector. Recently, Moreira et al. [14] developed a molecular assay based on real-time PCR to quantify *T. cruzi* load in triatomine samples. This methodology allows the study of different approaches related to the vector, such as vectorial capacity, including the ability of the triatomine to transfer the parasite during a blood meal. This approach can provide essential knowledge enabling understanding of the possible limiting factors for the development of different *T. cruzi* lineages [15,16], and also epidemiological analysis and tracking of field-collected triatomines, as recently reported for *Triatoma brasiliensis* in an acute ChD outbreak area [17].

*Triatoma brasiliensis* is a species complex comprising eight different members, each with different epidemiological importance, ecological requirements, morphological and genetic characteristics, and dispersion abilities [18,19]. In general, these taxa are, to varying extents, distributed in nine of the states of Brazil, with the occurrence of different species overlapping in some states [20,21,22]. This species complex is comprised of two subspecies and six species: *T. brasiliensis brasiliensis* Neiva, 1911; *T. b. macromelasoma* Galvão, 1956; *T. lenti* Sherlock & Serafim, 1967; *T. Juazeirense* Costa & Felix, 2007; *T. sherlocki* Papa et al. 2002; *T. petrocchiae* Pinto & Barreto, 1925; *T. bahiensis* Sherlock & Serafim, 1967; and *T. melanica* Neiva & Lent, 1941.

While *T. b. brasiliensis* is considered the main vector of public health importance in the semiarid region of Brazil, with numerous reports showing high rates of household (i.e., domestic and peridomestic) infestation and natural infection by *T. cruzi* [17,23,24,25,26], *Triatoma melanica* is considered exclusively sylvatic [27]. However, *Triatoma sherlocki,* also a member of the *brasiliensis* complex, was also previously considered exclusively sylvatic until it was found infesting human domiciles in a recently colonized community [28].

*Triatoma melanica* is rarely collected and poorly studied. This triatomine species are found in the Cerrado biome and is confined to the northwest of the state of Minas Gerais and the south of the state of Bahia [21,22,29]. *Triatoma melanica* is generally found in rocky outcrops in sylvatic environments, but it is also able to sporadically invade domiciles [27]. Until now, this species has only been reported to have a low *T. cruzi* infection rate, but it was considered competent to transmit the parasite due to the short time required for defecation after blood-feeding [30].

However, few studies have been conducted on the biological characteristics of this triatomine species. Sampling efforts have mostly focused on triatomine species that can colonize households, and there are few studies exclusively about sylvatic vectors. Knowledge about triatomine species acting as sylvatic vectors, and the possible risk of their transmitting *T. cruzi* to humans, is lacking. Hence, information on parasite load, the *T. cruzi* lineage infection, and the blood meal sources of the sylvatic triatomines in endemic ChD regions may provide important contributions toward understanding the contribution of *T. melanica* to *T. cruzi* transmission cycles in the field.

## 2. Materials and Methods

### 2.1. Field-Collected Triatomines

The current study was conducted in two municipalities in the north of the state of Minas Gerais: Espinosa (14°54′29″ S, 42°48′37″ W) and Monte Azul (15°09’18” S, 42°52’30” W). The distance between both municipalities is ~32 km. All sample sites were within the biogeographic zone characterized as the transition area between the Cerrado and the Caatinga biomes [31]. The insects were collected among rocky outcrops crevices using scissors and tweezers. Taxonomic identification was based on morphological diagnosis, following Dale et al. [29], as well as geometric morphometrics and DNA barcoding [22]. The sites of triatomine capture were recorded using a handheld geopositioning system to create a georeferenced map (Figure 1).

### 2.2. T. cruzi Infection by Optical Microscopy and In Vitro Cultivation

For parasitological analysis, we obtained triatomine faeces by abdominal compression, depositing the sample on a slide with saline solution, which was then covered with a glass coverslip. We examined the samples using a binocular optical microscope (OM) with 400× magnification, screening all fields for *T. cruzi*. In addition, all homogenates of the abdominal contents positive by OM for *T. cruzi* were directly cultivated in liver infusion tryptose medium (LIT), supplemented with 10% fetal bovine serum, and incubated at 38 °C [32]. Screening of cultures was performed using standard light microscopy during the first seven days of incubation. If positive, aliquots of the *T. cruzi* cultures were frozen at −20 °C until used for DNA extraction. Based on the visualization of the blood meal during dissection, we discarded starving specimens.

## 3. Molecular Assays

### 3.1. DNA Extraction

The intestinal content of each *T. melanica* specimen was macerated using a sterile crusher, and genomic DNA was isolated using the DNeasy Qiagen^®^ kit, according to the manufacturer’s protocol. In addition, DNA was extracted from the positive in vitro cultures. For some specimens, we obtained DNA samples both directly from the abdominal contents and indirectly from the in vitro cultures, enabling comparison of the genotyping results of the two different matched DNA samples derived from the same individual specimens. Quantification of the extracted DNA samples was determined using a NanoDrop™ One Microvolume UV-Vis Spectrophotometer (Thermo Scientific, Waltham, MA, USA). All DNA samples were stored at −80 °C until amplification by polymerase chain reaction (PCR).

### 3.2. T. cruzi Natural Infection by PCR

For comparative purposes, DNA samples (*n* = 67) from the abdominal contents of *T. melanica* were submitted to PCR for *T. cruzi* diagnosis. The set of primers used in this technique was the forward 121 (5′-AAATAATGTACGGG(T/G)GAGATGCATGA-3′) and the reverse 122 (5′-GGTTCGATTGGGGTTGGTGAATATA-3′) designed by Sturm et al. [33] and Wincker et al. [34]. The PCR reactions were performed with a final volume of 12 μL, containing: 10× Taq DNA polymerase buffer, 0.2 mM dNTPs, 3.5 mM MgCl_2_, 10 pmoles of each primer, 1U of Platinum *Taq* DNA polymerase (Invitrogen Life Technologies, Carlsbad, CA, USA), and 2 μL of DNA template. The thermocycling conditions consisted of an initial cycle at 95 °C for 5 min, followed by 35 cycles of denaturation at 95 °C for 45 s, annealing at 65 °C for 45 s, and extension at 72 °C for 45 s, with a final extension of 10 min at 72 °C. PCR amplification reactions were performed using a Veriti™ 96-well thermal cycler (AB Applied Biosystems, Foster City, CA, USA). The resulting PCR reactions were loaded onto a 6% polyacrylamide gel, run at 80 V in Tris-borate ethylenediaminetetraacetic acid running buffer, and subsequently stained with silver nitrate. When positive for *T. cruzi*, a fragment of 330 bp long was amplified, whereas 330 bp and 760 bp fragments indicated *Trypanosoma rangeli* infection [14]. All PCR reactions were run with three positive controls–two for *T. cruzi* (Colombian and Y strains), one for *T. rangeli* (Macias strain)–and a negative control (i.e., reaction mix, but without DNA template).

### 3.3. Genotyping of T. cruzi DTUs by Multilocus Conventional PCR

The methodology for *T. cruzi* genotyping was based on the analysis of *T. cruzi* SL-IRac, SL-IRI and II, 24Sα rDNA, and A10 targets by conventional multilocus PCR. Some assays were touchdown and hemi-nested PCRs to increase the specificity and sensitivity of *T. cruzi* genotyping, respectively [35]. This method has been proven to be able to detect all known DTUs, as well as mixed infections. The PCR reaction was performed according to the protocols published by the authors. PCR products were electrophoresed on 6% polyacrylamide gels and stained with silver nitrate. The identification of *T. cruzi* DTUs was performed following the flowchart presented by Ramirez and Moreira [34].

### 3.4. T. cruzi Quantification by Real-Time PCR (qPCR)

Quantification of the parasitic load was performed according to a methodology previously proposed by Moreira et al. [14]. The qPCR assays were performed using the multiplex Taqman system targeting the *T. cruzi* nuclear satellite DNA (sat-DNA) [36] and the mitochondrial 12S rRNA gene of triatomines [14]. The reaction mixture consisted of 2 μL DNA, 2×FastStart Universal Probe Master Mix (Roche Diagnostics GmbHCorp, Mannheim, Germany), 600 nM of the Cruzi1/Cruzi2 primers, and 250 nM of the Cruzi3 probe (FAM/NFQ-MGB), and 300 nM of the P2B primer, 500 nM of the P6R primer and 150 nM of the Triat probe (VIC/NFQ-MGB) (Applied Biosystems). Cycling conditions were 50 °C for 2 min, 95 °C for 10 min, followed by 45 cycles of 95 °C for 15 s and 58 °C for 1 min.

### 3.5. Identification of the Blood Meal Sources

In order to identify the host sources of triatomine blood meals, genomic DNA was subjected to PCR using a primer pair designed to bind conserved regions of the vertebrate 12S rRNA locus (L1085: 5′-CCCAAACTGGGATTAGATACCC-3′; and H11259: 5′-GTTTGCTGAAGATGGCGGTA 3′) [37]. Amplicons were purified using the Wizard^®^ SV gel and PCR Clean-Up System kit (Promega, Madison, WI, USA) and sequenced in both directions using the PCR primers. DNA sequencing reactions were performed using the BigDye^®^ Terminator v.3.1 Cycle Sequencing Kit (Applied Biosystems) and run on an ABI 3730 Sequencer. Bi-directional sequences were assembled. The resulting Sanger reads were edited using SeqMan (DNASTAR software package, DNASTAR Inc., Madison, WI, USA), and the consensus sequences generated were submitted to GenBank under Accession Numbers OP699675-OP699712. Our new sequences were compared with previously published sequences using BLASTN hosted on the NCBI server (http://www.ncbi.nlm.nih.gov/BLAST) (Accessed on 12 October 2022).

### 3.6. Statistical Analysis

Statistical analysis was performed using R [38] and the graphical user interface RStudio [39]. We compared, using ANOVA and Tukey’s post hoc test, the log-transformed parasite loads between both different vertebrate blood meal sources and *T. cruzi* lineages identified in triatomine vectors (that is, TcI, TcII, TcIII, and mixed infections). We examined the normality and homoscedasticity of the residuals using the Shapiro-Wilk and Levene’s tests, respectively. We also used a Pearson’s Chi-squared test with Bonferroni *post hoc* analysis for comparison of the TcI and TcII lineage frequencies between different vertebrate blood meal sources. In order to compare the two different methods (i.e., morphological and molecular) of parasite detection, we also analyzed the *T. cruzi* log-transformed parasite load in samples that were either positive or negative by optical microscopy using logistic regression analysis with the following model: $logit\left(Y\right)=\alpha +\beta \frac{1}{X}$, where *Y* was the diagnostic binary outcome optical microscopy, α the intercept, *β* the regression coefficient, and *X* the log-transformed parasite load as determined by qPCR. We calculated the goodness-of-fit using the pseudo-R-squared estimate following the method by Nagelkerke [40]. The difference in the log-transformed parasite loads between the different developmental stages of *T. melanica* was compared using ANOVA and Tukey’s post hoc test. To quantify the concordance between the results of different methods, generalized Kappa (hat k) coefficients were estimated according to the guide proposed by Landis and Koch [41].

## 4. Results

### 4.1. Insect Capture

A total of 141 *T. melanica* were captured in rocky outcrops in the municipalities of Espinosa and Monte Azul. We found all development stages, comprising 5 N1, 12 N2, 20 N3, 20 N4, and 24 N5 instar nymphs, as well as 32 adult males and 28 females.

### 4.2. T. cruzi Natural Infection Detected by Optical Microscopy and In Vitro Cultivation

Using OM, we were able to examine fecal samples from a total of 111 of the 141 specimens collected, of which 61 (~55%) were positive for *T. cruzi*-like parasites. Of these 61 *T. melanica* specimens positive for *T. cruzi* by OM, we successfully obtained axenic in vitro parasite cultures from 31 of them.

### 4.3. T. cruzi Natural Infection Detected by PCR

For the molecular assays, DNA samples extracted directly from the abdominal contents of 69 of the 141 insects captured were used. These samples were derived from five of the six developmental stages captured (i.e., N2 to adult), as well as both sexes. In addition, we also analyzed DNA samples derived from the 31 in vitro cultures of the abdominal contents from the *T. melanica* positive for *T. cruzi*-like parasites by OM. Overall, for 20 individual triatomine specimens, the DNA samples analyzed came from both the abdominal contents themselves as well as their paired in vitro culture, while for 11 individual triatomine specimens, the only DNA samples analyzed were from in vitro culture.

All 69 DNA samples from abdominal contents were tested using the kDNA-PCR, with 63 (~91%) of these specimens positive for *T. cruzi* by this method. Analysis of the DNA from in vitro cultures using the same kDNA-PCR method confirmed, in all instances, the presence of *T. cruzi* infection and discounted the possibility of the trypanosomes present or detected by OM being *T. rangeli*.

### 4.4. Comparison of T. cruzi Detection in T. melanica Using OM and Molecular Methods

A comparison of the 54 samples analyzed by both OM and kDNA-PCR demonstrated that 37 (~68%) and 49 (~91%) were positive for *T. cruzi*, respectively. Pearson’s Chi-squared test indicated that the difference between the results of these two methods was significant (χ^2^ = 8.22, *p*-value = 0.004).

### 4.5. Genotyping of T. cruzi DTUs

Out of the 63 kDNA-PCR-positive abdominal contents samples derived from *T. melanica,* the DTUs of the *T. cruzi* parasites of 56 (~89%) of these insects were genotyped. TcI (~55%, 31/56) was the most prevalent genotype, followed by TcIII (~20%, 11/56), and then only four individuals with TcII (~7%, 4/56). Mixed infections were detected in 10 (~18%) of the specimens, but only the TcI+TcIII combination was found. Regarding the 31 in vitro culture samples, 27 (~87%) were identified from the *T. cruzi* lineage, corresponding to 14 (~52%) infected by TcI, nine (~33%) by TcIII, and one by TcII (~4%). Considering mixed infections, only TcI+TcIII (~11%, 3/27) was detected.

### 4.6. Comparison of DTUs in Abdominal Contents and Culture Samples

As mentioned above, for 20 individual *T. melanica* specimens, we obtained DNA samples from both directly from the abdominal contents themselves as well as in vitro cultures derived from the former in order to compare the DTUs found in these paired DNA samples. We successfully genotyped 16 samples derived from abdominal contents and also by in vitro culture. Of these, 10 (~62%) specimens showed fully concordant results, with the same DTUs found in both DNA samples derived from the same individual *T. melanica*. Overall, the Kappa coefficient indicated moderate agreement between the *T. cruzi* DTUs present in the two different DNA sample types (κ = 0.429, *n* = 16, *z* = 2.66, *p*-value = 0.007).

### 4.7. Quantification of T. cruzi Parasite Load by Real-Time PCR (qPCR)

qPCR was performed on the 63 specimens of *T. melanica* that were positive by the kDNA-PCR, of which we successfully quantified the parasite load in 59 of these specimens. Of these, five (8.5%) were N2, 7 (12%) N3, 12 (20%) N4, eight (13.5%) N5 instar nymphs, and 14 (24%) were females and 13 (22%) males. The dynamic range of our qPCR assay was from 1 to 10^6^ parasite equivalents and from 0.0001 to 1 intestine equivalents. The observed dynamic range provided linear quantification over a 4-log and 6-log range for *T. cruzi* and triatomines, respectively, allowing an accurate standardization of the parasite loads. PCR efficiencies were 86.5% for the *T. cruzi* sat-DNA target and 83.2% for the triatomine 12S rRNA target. Also, the linearity coefficients (*R*^2^) were 0.99 for both targets. A wide range of the parasite loads in triatomines was observed, varying from 0.13 to 2.2 × 10^10^
*T. cruzi* per intestine. The observed median value for parasite load was 4.8 × 10^4^
*T. cruzi*/intestine equivalent (Figure 1).

### 4.8. Identification of Blood Meal Sources

We attempted to identify the blood meal source of 41 field-collected *T. melanica*, of which the majority were adult males (~21%) and N5 nymphal instars (~21%), with adult females (~18%) and N4 (~18%), N3 (~16%) and N2 (~6%) instars, also assayed. However, three mixed blood meals were observed, characterized by double peaks in the chromatograms, complicating the identification of the blood meal sources in these individuals by the DNA-sequence-based method used. The 12S rRNA sequences generated from the 38 individuals showed high identity (98–100%) to sequences available in the NCBI database. The sequences from three of these latter individuals (GenBank Accession Numbers OP699684, OP699690, and OP699706) all returned a BLAST identity of 97% with the same sequence in GenBank (MN206976.1). This latter sequence corresponds to a carnivore of the Mustelidae family, named *Mustela sibirita*, which does not occur in Brazil, suggesting that three of the *T. melanica* blood meals analyzed came from a close relative of this mustelid. The only member of the mustelid group that is known to occur in our study area is *Galactis cuja*, popularly known as the “furão” (or lesser grison), which has been previously reported in conservation units in Minas Gerais [42,43]. Thus, it seems likely that three of the *T. melanica* that we collected blood-fed on the mustelid *G. cuja*. In addition, four other species of vertebrate were detected as blood meal sources for *T. melanica*, of which 22 (~58%) were *Kerodon rupestris*, seven (~18%) *Thrichomys apereoides*, three (~8%) *Wiedomys cerradensis* and three (~8%) *Gallus gallus*.

### 4.9. T. cruzi Parasite Load According to DTUs and Blood Meal Source

When comparing *T. cruzi* parasite loads between *T. melanica* infected with different DTUs, TcII had significantly lower parasite loads than either TcI or TcIII (*F* value _[3,47]_ = 3.122, *p*-value = 0.0347) (Figure 2). However, no statistical difference was found between mixed (TcI+TcIII) and single infections. Also, no significant differences were found between parasite load and different blood meal sources (*F* value _[4,31]_ = 0.688, *p*-value = 0.60) (Figure 3).

### 4.10. T. cruzi Parasite Load According to Triatomine Developmental Stage

We found a significant difference when comparing the parasite loads between different developmental stages of *T. melanica*, but this was significant only when comparing adult females with either N2, N4, and N5 nymphal instars, with parasite loads higher in adult females (*F* value _[5,50]_ = 3.653, *p*-value = 0.006) (Figure 4).

### 4.11. Parasite Load According to T. cruzi Positivity Using OM

The regression coefficient estimates were statistically significant, and the estimates for α (intercept) and (regression coefficient) were 5.16 ± 2.13 S.E. (*Z* = 2.412, *p*-value = 0.015) and −17.70 ± 8.97 S.E. (*Z* = −1.97, *p*-value = 0.048), respectively. The model had a pseudo-*R*^2^ of 0.16. Parasite load had a significant effect on the outcome of OM. Thus, specimens with higher parasite loads had a greater probability of being diagnosed as *T. cruzi*-positive using OM. According to the logistic regression model, the odds ratio for the *T. cruzi*-positive OM diagnosis was approximately 29.1 when the parasite load was 10^7^ compared to 10^3^. Data for parasite loads less than 10^3^ were excluded from this analysis, as there were too few observations for this range (Figure 5).

## 5. Discussion

Northern Minas Gerais is an important endemic area for ChD in Brazil [44], sharing a boundary with the Brazilian northeast, one of the poorest regions in the country, which is classified as an underdeveloped region. Recently, the municipality of Espinosa was reported by the Minas Gerais State Health Service as an important site where ChD could be reemerging because of the high prevalence of patients with chronic ChD [45]. Historically, this region has high rates of *T. infestans* and *T. sordida* infestation [46]. However, other triatomine species also occur in this area, such as *T. melanica*. Despite the fact that this latter species is still considered exclusively sylvatic, adult specimens are often found in houses in this region [47].

The present study employed a molecular epidemiological approach in two areas of the state of Minas Gerais, Brazil. To date, this is the first report on the parasite load of naturally infected *T. melanica* and its correlation with different *T. cruzi* genotypes, blood meal sources, and results of OM diagnosis. The most important findings were: (i) TcI is the predominant *T. cruzi* lineage in the study region; (ii) the genotyping results from both intestine samples and in vitro culture had moderate agreement; (iii) rodents were the main source of *T. melanica* blood meals, in particular, the rocky cavy *K. rupestris* was the most commonly identified host; (iv) in general, *T. melanica* had high *T. cruzi* loads, but females had higher parasite loads than N2, N4, and N5 nymphal instars; (v) the diagnostic positivity for *T. cruzi* differed significantly between the OM and kDNA-PCR methods; (vi) the probability of detecting *T. cruzi* by OM increased at higher parasite loads.

Herein, we detected a wide range of parasite loads in the field population of *T. melanica*, varying from 10^−1^ to 10^10^
*T. cruzi* per intestine, as already observed for field-captured triatomines [14,17]. Recently, Saavedra et al. [48] also determined *T. cruzi* loads in the sylvatic vector, *Mepraia spinolai*, collected in the field in Chile, but the median value did not exceed 10^2^ parasites per insect. The observed median value for parasite load in the present study was 10^4^, 10 times lower than that reported in *T. brasiliensis* during an orally transmitted ChD outbreak area in northeastern Brazil [17]. The latter authors suggested that native rodents, potential *T. cruzi* reservoirs, could be the link between domestic and sylvatic *T. cruzi* transmission cycles, resulting in a high parasitic load in *T. brasiliensis*. Although both *T. melanica* and *T. brasiliensis* differ in their ability to colonize artificial ecotopes, the environmental spaces currently occupied by these two triatomine species are similar, involving rocky outcrops [22], and these latter natural ecotopes are frequently inhabited by rodents [49,50].

Although no significant difference was observed in parasite load according to blood meal source, all *T. melanica* specimens analyzed, especially those that had blood-fed on rodents (~87% of individuals), had high levels of parasites, exceeding 10^4^
*T. cruzi* per intestine. The cavy *K. rupestris* was the main blood meal source detected, as similarly previously observed for *T. brasiliensis* in the same north-eastern semiarid area. This vertebrate species is considered one of the key blood meal sources for triatomines and a reservoir host of *T. cruzi* [50]. Our observations suggest that this rodent could also be a potential *T. cruzi* reservoir for *T. melanica* in this area. It is important to highlight that *K. rupestris* is a well-known reservoir host of the *T. cruzi* TcI lineage [51,52] and that this DTU was the predominant lineage found in the present study. In general, TcI is the most prevalent genotype in the Americas and most frequently identified in sylvatic cycles [11]. Previously, Wanieck et al. [53] only found TcI in *T. melanica*, although the sample size analyzed was small. Moreover, this DTU is the most predominant in the order Rodentia [11], suggesting that the host and vector participate in the same *T. cruzi* transmission cycle in our study area. Since rodents such as *K. rupestris* and *Thrichomys* sp. can invade households, sheltering in peridomestic structures, and play a major role as blood meal sources, triatomines follow them into domestic and peridomestic environments [24,50,54,55]. This scenario has significant implications for understanding the potential epidemiological importance of sylvatic populations of *T. melanica* in the transmission of *T. cruzi*.

The TcIII genotype was the second most identified *T. cruzi* lineage in *T. melanica*, followed by the mixed infection TcI + TcIII, with TcII having the lowest prevalence of any of the DTUs detected. Previously, TcIII was significantly more frequently detected in sylvatic cycles than in domestic cycles [11] and was associated with sylvatic vectors [56,57], consistent with our findings reported here. However, the DTU TcIII has also been recorded in human cases of ChD in Brazil [11,58,59,60,61], which could be a cause for concern because of the proximity between the sylvatic transmission cycle and humans in this area and the sporadic cases of domicile invasion by adult triatomines. It is important to highlight that TcIII may be under-reported in both domestic and sylvatic transmission cycles because some typing methodologies fail to distinguish between TcIII and TcIV [62]. Regarding the TcII, this genotype is more frequently identified in domestic cycles [9,11], consistent with the low infection prevalence of this DTU observed in our study. Also, this DTU had lower parasite loads compared to the other DTUs detected. According to Araújo et al. [63], experimental infection of the TcI and TcII isolates in *T. brasiliensis* have different patterns of development in the digestive tract of this triatomine species: the genotype TcI colonized the intestine predominantly, in contrast to TcII. It is known that evolutionary pressure on parasite development in vectors may determine selection on subpopulations [1], in addition to the presence of certain bacteria species in the gut microbiome of triatomines, which inhibit the growth of some *T. cruzi* genotypes [15].

We did not find any association between the occurrence of specific DTUs and blood meal source, nor between parasite load and any of the DTUs detected. However, when we analyzed the difference among nymphal instars, females had higher parasite loads than either N2, N4, or 5N instar nymphs. There is a consensus that blood feeding has a direct effect on the reproductive efficiency of triatomines since blood meals stimulate endocrine regulation of ovarian development and is necessary for egg production [64]. Furthermore, it was reported that *T. melanica* needs many blood meals to molt [30]. Thus, it is possible that females tend to be reinfected more frequently during their life cycle, increasing their *T. cruzi* parasitic load. Also, chickens were identified as the blood meal source in three specimens collected from two sites. The latter is not an unexpected finding since the rocky outcrops where these individuals were collected were near a domicile (~10 m) containing a henhouse, and all the insects collected that had blood-fed on chickens were adults, consistent with their migration by flight.

In this study, we also reported a moderate agreement between the DTU genotyping results performed using DNA extracted directly from gut samples or in vitro cultures of the latter. This finding goes against the common, widespread idea among the scientific community that in vitro culture limits the analysis of genotypic diversity due to selective pressures during cultivation, suppressing some populations. However, it is relevant to note that the culture samples were inoculated to a new medium only three times (~15 days), which may have avoided the selection of genotypes. Hence, this result must be interpreted with caution.

Additionally, we observed here a significant difference between different *T. cruzi* detection methods used to evaluate infection in triatomines. Although most surveillance and control programs for ChD have technicians trained in identifying *T. cruzi* infection using OM, it is well-known that the detection can be underestimated by this methodology [14,65,66]. There are some reports showing the difference between both methods. According to Moreira et al. [14], the kDNA-based PCR screening of field-collected triatomines was shown to be more sensitive than a microscopic examination with a positivity of 21%, whereas only 7% of positive PCR results were confirmed by OM of fecal drops of live insects. In another investigation, 10.7% of *T. sordida* specimens captured in Mato Grosso do Sul were positive for flagellated protozoa, as determined by microscopic search, and 18.1% were positive for *T. cruzi* using kDNA-PCR [67]. It is important to highlight that molecular analysis was performed with DNA obtained from the midgut and rectum of insects. Thus, it could, in part, explain the differences in positivity between microscopy and PCR, considering the lower number of parasites in triatomine-diluted feces when compared with intestinal content homogenates used for DNA extraction, and thus yielding false negative results by the traditional OM diagnosis. Also, the nutritional status of the bugs at the time of capture, and the examination of dead insects, can make an accurate estimation of the prevalence of *T. cruzi* difficult [65], in addition to the difficulty of morphologically distinguishing between *T. cruzi* and *T. rangeli*. The latter trypanosome species has already been reported to occur in the *Triatoma* genus [17,68].

Moreover, it seems coherent that the higher the parasite load, consequently the higher possibility of detecting positive results by OM. We estimated the odds ratio of detecting positive tests according to the parasite load. When the *T. cruzi* load was 10^7^ parasites per intestine, the chance of detecting a real positive result is ~29 times higher compared to a parasitic load of 10^3^. However, in samples with 10^3^
*T. cruzi*/intestine, the probability of being detected was 30%, consequently resulting in a 70% false-negative, which could bias studies on the prevalence and vigilance of *T. cruzi* infection. The tracking of pathogen presence in vectors allows epidemiologists to evaluate transmission risks in time and space, being crucial for strategy designs for disease prevention [66]. The examination of *T. cruzi* infection in triatomines is, therefore, an important component of the Chagas disease control program [69]. Unfortunately, we must consider the reality of entomological-parasitological routine surveillance, which, most of the time, does not have the adequate structure and equipment to carry out molecular techniques. In a short time, a viable solution to this problem is to employ statistical methods to minimize the error. There are some methodological strategies that allow the computation of corrected estimates of infection frequency in triatomines, helping enhance vector-borne disease surveillance systems when pathogen detection is imperfect [66].

## 6. Conclusions

Our results presented here are the most comprehensive study to date on *T. cruzi* parasite load, *T. cruzi* DTU genotyping, and blood meal source identification in field-collected *T. melanica.* We observed a wide range of variations in the parasite loads of this triatomine species, with a median value of 10^4^ parasites/intestine. Triatomines fed on rodents, which were the most frequently detected blood meal source in the current study. *Kerodon rupestris* was the main blood meal source identified, suggesting it is a *T. cruzi* reservoir host within our study region. Adult female *T. melanica* had higher parasite loads, indicating its potential epidemiological importance in *T. cruzi* transmission and the need for control interventions to pay more attention to this triatomine species since adult specimens sporadically invade human dwellings in the study region. The predominant genotype in single infections was TcI, followed by TcIII. Also, we compared *T. cruzi* detection methods (kDNA-PCR and OM) and observed that the probability of detecting true-positives by OM was low when the parasite load was low. In conclusion, the multidimensional approach used here contributes to our knowledge of *T. cruzi* diagnosis, as well as biological and epidemiological aspects of the sylvatic population of *T. melanica* in the study region, highlighting the need to extend surveillance and control to this vector to prevent new cases of ChD.

## Figures and Tables

**Figure 1 pathogens-11-01498-f001:**
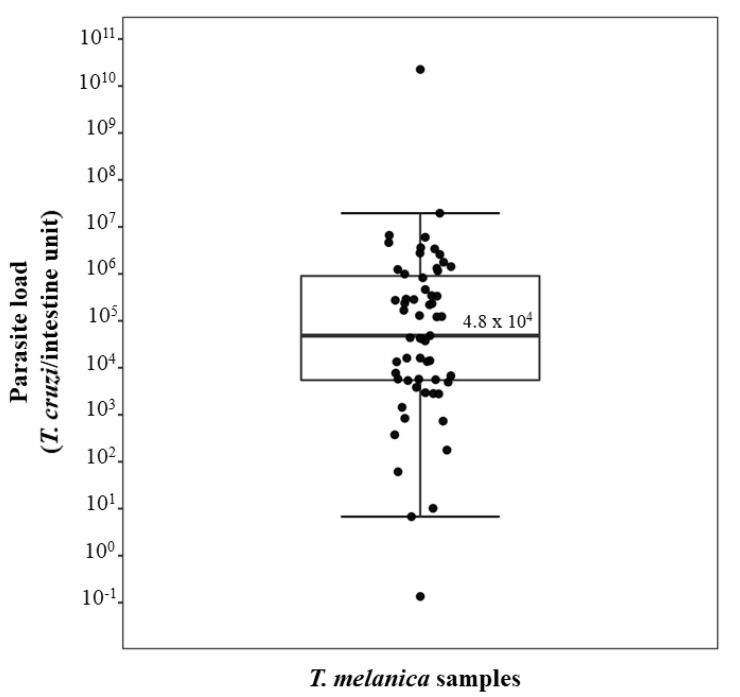
Box plot showing the parasite load in 59 *T. melanica* specimens collected in the municipalities of Espinosa and Monte Azul. The box illustrates the median and interquartile range, while the whiskers give the “minimum” and “maximum” values, and the open circles represent outliers.

**Figure 2 pathogens-11-01498-f002:**
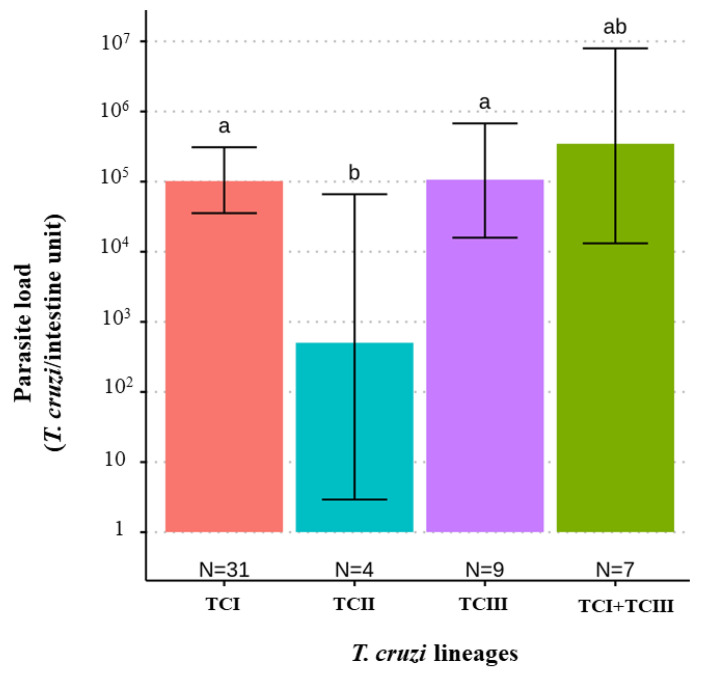
Bar plot comparing estimates of the *T. cruzi* parasite load (95% C.I.) according to the infecting *T. cruzi* DTU lineage in 51 *T. melanica* from the municipalities of Espinosa and Monte Azul. Groups with different letters indicate significant difference (*p* < 0.05). Groups not sharing any letter are significantly different.

**Figure 3 pathogens-11-01498-f003:**
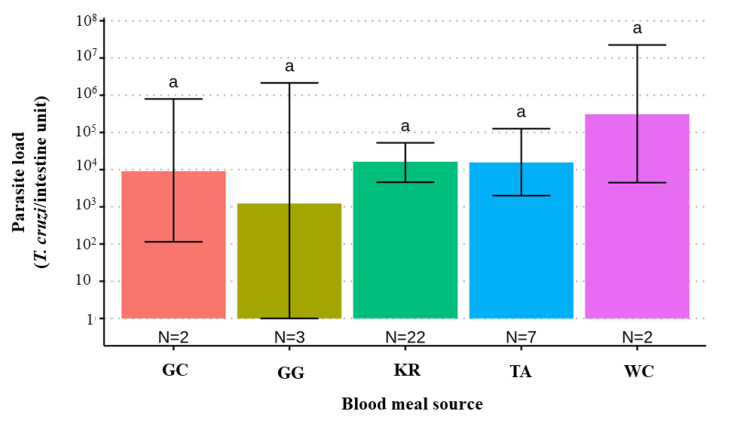
Bar plot comparing estimates of the *T. cruzi* parasite load (95% C.I.) according to blood meal source in 51 *T. melanica* from the municipalities of Espinosa and Monte Azul. GC: *Galactis cuja*; GG: *Gallus gallus*; KR: *Kerodon rupestris*; TA: *Thrichomys apereoides*; WC: *Wiedomys cerradensis* Groups with different letters indicate significant difference (*p* < 0.05). Groups not sharing any letter are significantly different.

**Figure 4 pathogens-11-01498-f004:**
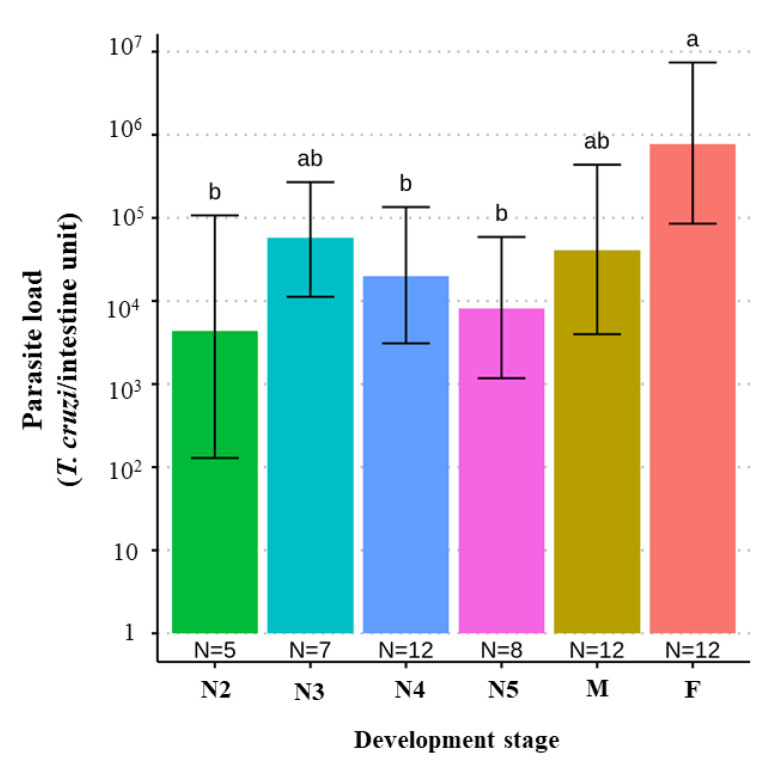
Bar plot comparing the estimate of the *T. cruzi* parasite load (95% C.I.) according to the development stages of 51 *T. melanica* from the municipalities of Espinosa and Monte Azul. Groups with different letters indicate significant difference (*p* < 0.05). Groups not sharing any letter are significantly different.

**Figure 5 pathogens-11-01498-f005:**
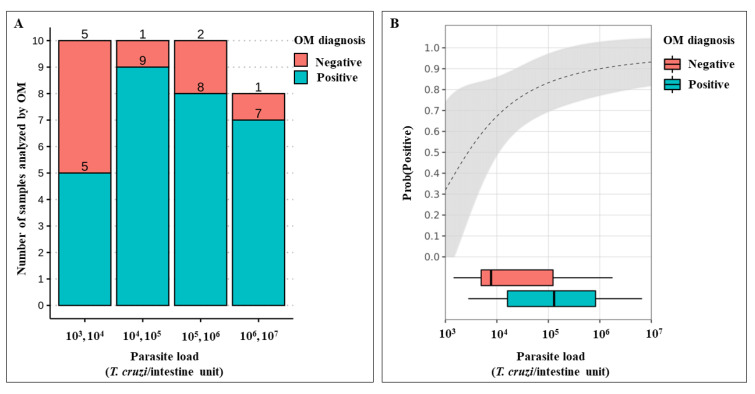
Association between *T. cruzi* diagnosis using optical microscopy (OM) and parasite load. (**A**) Stacked histogram showing the number of *T. melanica* specimens diagnosed by OM as either negative or positive for *T. cruzi* according to their parasite load. (**B**) Positive diagnosis probability estimate given the parasite load (the grey area gives 95% C.I. of this estimate).

## Data Availability

Data are contained within the article. Sequence data were deposited in GenBank.

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
