# Peer review of "High Parasitic Loads Quantified in Sylvatic Triatoma melanica, a Chagas Disease Vector"

_pathogens, 2022, doi:10.3390/pathogens11121498_

Round 1

Reviewer 1 Report

This manuscript, High parasitic loads quantified in sylvatic Triatoma melanica, a 2

Chagas disease vector by Barbosa et al., details the current standing of several T cruzi discrete tying units by their effective presence and abundance in the insects vectors of the complex Triatoma brasiliensis surrounding the rural regions in the state of Minas Gerais. This is a very well written manuscript and its results are important for disease diagnostics and surveillance.

Line 36. Sentence is very long and confusing.

Line 247. The difference between the results is significant? Why so large? Justifies use of kDNA

Line 256. Typeo, were identified [from] the T. cruzi lineage

Line 263 why is about 50/50 that some are concordant and the others not

Line 282 a box plot with individual data points would be more informative

Line 328, were there any difference when nymphs were separated by sex?

Line 342 what about by kDNA? If the importance is to demonstrate how OM can be insufficient visualizing the difference as described in 247 would be a good justification for different diagnostic tools

Line 426 typeo, we chickens

Line 443 discuss change in diagnostic routines from OM to …

Author Response

Dear Editor,

We would like to thank the reviewers for their careful revision of this manuscript. We accept all recommendations, which were important to improve our study. Below, you can see that we answered (in red) all questions point by point. There were just a few points that we could not follow, which are justified accordingly. See doc attached. 

Reviewer 2 Report

The article entitled “High parasitic loads quantified in sylvatic Triatoma melanica, a Chagas disease vector” presents important results of a research on Triatoma melanica as a vector of Trypanosoma cruzi in two municipalities in Brazil.  The study establishes and presents conclusive data on the parasitic loads of the vector, the predominant discrete tying units of T. cruzi found by genotyping in the vector, and the reservoir animal hosts that most often are source of a blood meal for T.melanica in the study area. In addition, the study compares the effectiveness of light microscopy and molecular techniques as diagnostic methods for detection of T. cruzi infection. 

However, I do have some minor remarks regarding the presentation of the material and its clarity:

ABSTRACT

From line 17 to line 20 – It is a total mess with these percentages.  Indeed, all of this is better explained in the results section, but the abstract should stand on its own. At a quick glance it is clear that 61 is not 55% from 141 and so on. Please rewrite and clarify.

Line 21- “… TcI+TcI mixed…”, Probably the authors mean TcI+TcIII? Please correct it.

RESULTS

From line 222 to line 224 – “A total of 141 T. melanica were captured in rocky outcrops in the municipalities of 222 Espinosa and Monte Azul. We found all development stages, comprising 32 N1, 12 N2, 20 223 N3, 20 N4, and 24 N5 instar nymphs, as well as 32 adult males and 28 females.” The total number I get is 168, not 141. Why stage N1 is mentioned at all, if nowhere in the rest of the text it is mentioned again? Please rewrite and clarify.

From line 250 to line 254 – Including the 10 mixed infections in the numbers of single infections again makes the math confusing. Please rewrite and clarify.

DISCUSSION

Line 426 – There is some terrible mechanical mistake with that chicken. Please correct it.

The article entitled “High parasitic loads quantified in sylvatic Triatoma melanica, a Chagas disease vector” presents important results of a research on Triatoma melanica as a vector of Trypanosoma cruzi in two municipalities in Brazil.  The study establishes and presents conclusive data on the parasitic loads of the vector, the predominant discrete tying units of T. cruzi found by genotyping in the vector, and the reservoir animal hosts that most often are source of a blood meal for T.melanica in the study area. In addition, the study compares the effectiveness of light microscopy and molecular techniques as diagnostic methods for detection of T. cruzi infection. 

However, I do have some minor remarks regarding the presentation of the material and its clarity:

ABSTRACT

From line 17 to line 20 – It is a total mess with these percentages.  Indeed, all of this is better explained in the results section, but the abstract should stand on its own. At a quick glance it is clear that 61 is not 55% from 141 and so on. Please rewrite and clarify.

Line 21- “… TcI+TcI mixed…”, Probably the authors mean TcI+TcIII? Please correct it.

RESULTS

From line 222 to line 224 – “A total of 141 T. melanica were captured in rocky outcrops in the municipalities of 222 Espinosa and Monte Azul. We found all development stages, comprising 32 N1, 12 N2, 20 223 N3, 20 N4, and 24 N5 instar nymphs, as well as 32 adult males and 28 females.” The total number I get is 168, not 141. Why stage N1 is mentioned at all, if nowhere in the rest of the text it is mentioned again? Please rewrite and clarify.

From line 250 to line 254 – Including the 10 mixed infections in the numbers of single infections again makes the math confusing. Please rewrite and clarify.

DISCUSSION

Line 426 – There is some terrible mechanical mistake with that chicken. Please correct it.

Author Response

Dear Editor,

We would like to thank the reviewers for their careful revision of this manuscript. We accept all recommendations, which were important to improve our study. Below, you can see that we answered (in red) all questions point by point. There were just a few points that we could not follow, which are justified accordingly. See doc attached!
